# A phenomenological universal expression for the condensate fraction in strongly-correlated two-dimensional Bose gases

G.E. Astrakharchik[1*], I.L. Kurbakov[2], N.A. Asriyan[3] and Yu. E. Lozovik[2]

**1** Departament de Física, Campus Nord B4-B5, Universitat Politècnica de Catalunya, E-08034 Barcelona, Spain
**2** Institute for Spectroscopy RAS, Troitsk 108840, Moscow, Russia
**3** N.L. Dukhov Research Institute of Automatics (VNIIA), Moscow 127030, Russia
* grigori.astrakharchik@upc.edu

August 28, 2025

## Abstract

We investigate the relation between non-local and energetic properties in 2D quantum systems of zero-temperature bosons. By analyzing numerous interaction potentials across densities spanning from perturbative to strongly correlated regime, we discover a novel high-precision quantum phenomenological universality: the condensate fraction can be expressed through kinetic energy and quantum energy, defined as total energy relative to classical crystal state. Quantum Monte Carlo simulations accurately validate our analytical expression. Furthermore, we test the obtained relation on the fundamental example of a non-perturbative system, namely, the liquid helium. The proposed relation is relevant to experiments with excitons in transition metal dichalcogenides (TMDC) materials, as well as ultracold atoms and other quantum systems in reduced dimensionality.

## 1  Introduction

The precise description of correlated many-body systems has been amongst the most challenging topics of physics in recent decades. Since their experimental realization [1–3], Bose-Einstein condensates in ultracold atoms became a prospective playground for investigating effects due to strong interactions. One of the most striking manifestations of quantum phenomena is occurrence of Bose-Einstein condensation in Bose systems at ultralow temperatures. In this phenomenon, the number of atoms in the condensate, denoted as $N_0$, becomes macroscopic and is proportional to the total number of particles $N$. The Gross-Pitaevskii equation describes the time evolution of the condensate wave function, which in a homogeneous system corresponds to a state with zero momentum, $k = 0$. The ground state energy in a fully condensed homogeneous system, $N_0/N = 1$, is given by the mean-field expression, $E/N = 2\pi\hbar^2 na/m$, where $n$ is the average density of atoms of mass $m$ and $a$ is the $s$-wave scattering length [4]. In a more precise treatment, weakly correlated systems can be described by Bogoliubov theory (further BT) [5] with perturbations given by means of the diagrammatic approach developed by Beliaev [6,7].

One of the experimentally accessible experimental signatures of a strongly correlated system is condensate depletion at $T = 0$. In ultracold atomic gases, the condensate fraction is typically measured using the time-of-flight (ToF) method, as a fraction of atoms that do not fly away after the trapping potential is released, and corresponds to the measurement of the momentum distribution for $k = 0$ momentum [8]. For weakly correlated system one may derive

an expression for the condensate fraction by in the scope of the Bogoliubov theory:

$$\frac{N_0}{N} = 1 - \frac{8}{3\sqrt{\pi}}\sqrt{na^3} + \cdots. \tag{1}$$

Along the same lines the correction to the mean-field energy [9] is given as

$$\frac{E_{3D}}{N} = \frac{2\pi\hbar^2 na}{m}\left[1 + \frac{128}{15\sqrt{\pi}}\sqrt{na^3} + \cdots\right]. \tag{2}$$

A comparison of Eqs (1-2) shows that the beyond-mean-field terms in the condensate fraction and the energy share the same functional form.

In two-dimensional (2D) geometry, both the condensate fraction

$$\frac{n_0}{n} = 1 - \frac{1}{|\ln(na^2)|} + o\left[\frac{1}{\ln(na^2)}\right] \tag{3}$$

and the energy per particle

$$\frac{E}{N} = \frac{2\pi\hbar^2 n}{m}\frac{1}{|\ln(na^2)|}\left[1 + o(1)\right] \tag{4}$$

have a weak (logarithmic) dependence on the gas parameter $na^2$ [10].

Notably, expansions (1)-(4) of the equation of state (EoS) in terms of the gas parameter are *universal* as they do not depend on microscopic details of interaction between bosons. Some of the non-universal higher-order terms have been investigated [11–13] and the subsequent perturbative terms have been obtained for the 2D [14–16] and 3D [17] cases.

When it comes to verification of the low-density expansions, Quantum Monte Carlo (QMC) methods have proven to be extremely useful both in 3D [11,18] and 2D [13,19] systems. In addition, the ability to tune the interparticle interactions via the magnetic Feshbach resonance made it possible to experimentally verify some of the theoretical predictions [11,20].

Despite the efforts to describe the systems with arbitrarily strong correlations, complete results are still missing due to complexity of the many-body system. On the other side, there has been a substantial increase both in the variety and the number of accessible Bose systems in recent years. In particular, experiments have been carried out with quantum well excitons [21–23], dipolar atoms and molecules in optical lattices [24,25], magnons [26], excitonic polaritons [27,28] as well as microcavity photons [29] along with the discovery of atomically thin transition metal dichalcogenides (TMDC) layers [30] which appeared to be also suitable for exploring Bose condensation [31,32]. Despite the variety of platforms, Bose-Condensates in different particle systems usually share universal features (see [33,34] for a discussion and a review on the topic). Theoretical description of the broad variety of up-to-date experiments requires developing methods which may be applied to strongly correlated systems. The goal of this study is to partly contribute to this goal using numerical evidence gained from Monte-Carlo simulations.

For a dilute two-dimensional Bose gas, one may combine the perturbative predictions (3) and (4) up to the lowest order in $1/\ln(na^2)$:

$$\frac{n_0}{n} = 1 - \frac{1}{2\pi}\frac{mE}{\hbar^2 Nn} = 1 - \frac{1}{2}\frac{E}{NE_F}. \tag{5}$$

Although the expressions for the condensate fraction and the energy are non-universal for strongly correlated gases, a universal relation between them may have a broader range of validity. As we demonstrate in this paper, it appears to be the case. We propose an extended

version of (5), a phenomenological universal relation that is reasonably well applicable in a broad range of densities for various functional forms of interparticle interaction potential.

Such phenomenological universalities are ubiquitous in physics. Besides the one mentioned above for weakly correlated Bose gases, numerous other relations have been identified. Among the recognized results are the universalities in metal EoS [35], the Lindemann melting criteria in 3D [36] and in 2D [37]. Additionally, novel results have been found regarding universal behavior of classical [38, 39] and quantum [40] particle systems as well as in the areas of astrophysics (e.g. universalities in neutron star deformabilities [41]) and polymer science (e.g. a universal EoS of solved natural polymers [42]).

The paper is organized as follows. In Section 2 we describe a 2D Bose gas model and provide a list of various interaction potentials under consideration. Then follow details of Monte-Carlo simulations. In Section 3 we present the raw results of simulations and analyze them to propose an implicit expression for the condensate fraction in terms of the so-called "quantum" energy and kinetic energy of the gas. Finally, we discuss the results in Section 4, where we provide evidence on the universal nature of the discovered relation and describe its applicability limits.

## 2 Model Hamiltonian and numerical methods

To demonstrate a universality in some quantity, it is necessary to establish that its value is independent of the specific details of the interaction potential. To do so, we introduce and examine multiple model Hamiltonians labeled by index $\ell$, each characterized by different interparticle interaction potential $U_\ell(r)$. The microscopic Hamiltonian of bosons of mass $m$ can be expressed in a general form as

$$\hat{\mathcal{H}}_\ell = -\frac{\hbar^2}{2m} \sum_{i=1}^{N} \Delta_i + \sum_{i<j} U_\ell(|\mathbf{r}_i - \mathbf{r}_j|) \,, \tag{6}$$

where $\mathbf{r}_i$ denotes the position of each of $N$ particles within the 2D simulation box with periodic boundary conditions. We have analyzed a total of 8 interaction potentials, each with unique characteristics. The details for $U_\ell(r)$ for each potential are summarized in Table 1. The specific choice of potentials is guided by two main considerations of including experimentally relevant potentials while also ensuring that they are physically significantly different. All of them fall off fast enough to be integrable at $r \to \infty$, which ensures the applicability of (3) for low density gaseous phase. The considered potentials include both short-range (*e.g.* hard core potential $U_1(r)$ typical for ultracold atoms) and long-range ones (dipolar atom interaction $U_2(r)$) as well as their combination ($U_3(r)$). We include the Lennard-Jones potential $U_4(r)$, which is probably the most extensively studied model potential for interatomic interactions. We also consider its combinations with a dipolar potential $U_5(r)$. The potential $U_6(r)$ models interaction between indirect excitons and provides, along with the renowned Yukawa potential $U_7(r)$, a rare experimentally relevant example of a potential with integrable singularity at $r \to 0$, i.e. with a "soft core". In addition, we consider a system of 2D helium atoms with interaction potential $U_8(r)$, which enables us to explore a Bose system in the vicinity of the liquid-gas phase transition.

### 2.1 Details of Monte Carlo simulation

For each of the model interaction potentials, we consider a system of $N$ particles in a square box of size $L \times L$ with periodic boundary conditions imposed. For performing a diffusion Monte

| $\ell$ | $U_\ell(r)$ | Description | Abbreviation | Parameters |
|---|---|---|---|---|
| 1 | $\begin{cases} \infty, & r < r_0, \\ 0, & r > r_0 \end{cases}$ | Hard core interaction | HC | — |
| 2 | $\dfrac{\hbar^2 r_0}{mr^3}$ | Dipole-dipole interaction in 2D | DD | — |
| 3 | $\dfrac{\hbar^2}{mr_0^2}\dfrac{r_0^k}{r^k}$ | Power-law short-range potential | SR | $k = 9$ |
| 4 | $\dfrac{\hbar^2 U_0}{mr_0^2}\left(\dfrac{r_0^{12}}{r^{12}} - \dfrac{r_0^6}{r^6}\right)$ | Lennard-Jones (LJ) potential | LJ | $U_0 = 4.8$ |
| 5 | $\dfrac{\hbar^2}{mr_0^2}\left(\dfrac{l^{12}}{r^{12}} - \dfrac{l^6}{r^6}\right) + \dfrac{\hbar^2 r_0}{mr^3}$ | LJ with dipolar repulsion | LJ+DD | $l = 2r_0$ |
| 6 | $\dfrac{2\hbar^2 U_0}{mr_0}\left(\dfrac{1}{r} - \dfrac{1}{\sqrt{r^2 + r_0^2}}\right)$ | Separated dipoles (*e.g.* indirect excitons) | SD | $U_0 = 4$ |
| 7 | $\dfrac{U_0}{mr_0 r}\exp\left(-\dfrac{r}{r_0}\right)$ | Yukawa potential | Yu | $U_0 = 90$ |
| 8 | Aziz2 potential [43] | Liquid helium | 2DHe | See table I in Ref. [43] |

Table 1: Details for different interaction potentials employed in our simulations. Here, $r_0$ denotes the unit of length used in Monte Carlo simulations. The values of the power-law exponent $k$, potential amplitude $U_0$, Lennard-Jones length $l$ are explicitly reported.

Carlo (DMC) calculation, we use the Jastrow form trial (guiding) function

$$\psi_T(\mathbf{r}_1, \cdots, \mathbf{r}_N) = \prod_{1 \leq j < k \leq N} f_T(|\mathbf{r}_j - \mathbf{r}_k|). \tag{7}$$

We construct the pair Jastrow function $f_T(r)$ by smoothly matching: (i) numerical solution of a two-particle scattering problem with positive scattering energy, $E_T > 0$, for distances $r < R_T$; (ii) asymptotic hydrodynamic behavior [44] for $R_T < r < L/2$; (iii) unit value, $f_T(r) = 1$, for $r \geq L/2$ (see also Eq. (2) in Ref. [45]). Variational parameters $R_T$ and $E_T$ are optimized by minimizing the variational energy. The size of the box is adjusted to reproduce the desired value of the density according to $n = N/L^2$.

We are mostly interested in the following quantities:

1. Ground state energy $E_\ell = \langle \hat{\mathcal{H}}_\ell \rangle$ obtained directly via a DMC calculation.

2. Condensate fraction $n_0/n$, which is evaluated from the static structural factor and extrapolated to $N \to \infty$ employing quantum hydrodynamics (see Sec. 6 in Ref. [40]).

3. Potential energy $E_\ell^{\text{pot}} = \left\langle \frac{1}{2} \sum_{i \neq j} U_\ell(|\mathbf{r}_i - \mathbf{r}_j|) \right\rangle$ and the kinetic one $E_\ell^{\text{kin}} = E_\ell - E_\ell^{\text{p}}$. Here $i$ enumerates all the particles inside the simulation box, while, due to periodic boundary conditions, $j$ runs over both the particles in the box and their images (including images of $i^{\text{th}}$ particle).

The total energy is computed using a pure estimator, while all other observables are evaluated via the standard extrapolation technique [46]. For the static structure factor, results obtained from the pure estimator agree with those from the extrapolation method within statistical uncertainties. Simulations were carried out for systems containing $N = 64$, 100 and 168. The results presented in the main text correspond to simulations with $N = 64$ particles.

## 3   Condensate fraction

In this section, we explore a possible connection between the condensate fraction and the energetic properties of the system at zero temperature. As the Bogoliubov perturbative expansion is limited to systems with weak interactions and small quantum depletion, we aim to extend the region of validity of the expression (5).

### 3.1   Constant-exponent relation based on the total energy

Assuming that expressions (3-4) represent the leading terms of a power series, and given that the condensate fraction $n_0/n$ is bounded to the interval $(0, 1)$, we propose a straightforward generalization of Eq. (5). Treating it as a series expansion of an exponent, we express the condensate fraction as an exponential function of the energy per particle $E/N$ measured in the unit of the characteristic energy $\hbar^2 n/m$,

$$\frac{n_0}{n} = \exp\left(-\frac{\varepsilon}{2\pi}\right), \tag{8}$$

with shorthand notation $\varepsilon = mE/(\hbar^2 N n)$ being introduced. In relation (8), the decay constant $(2\pi)$ provides the characteristic value of the dimensionless energy per particle $\varepsilon$ for which the condensate becomes significantly depleted.

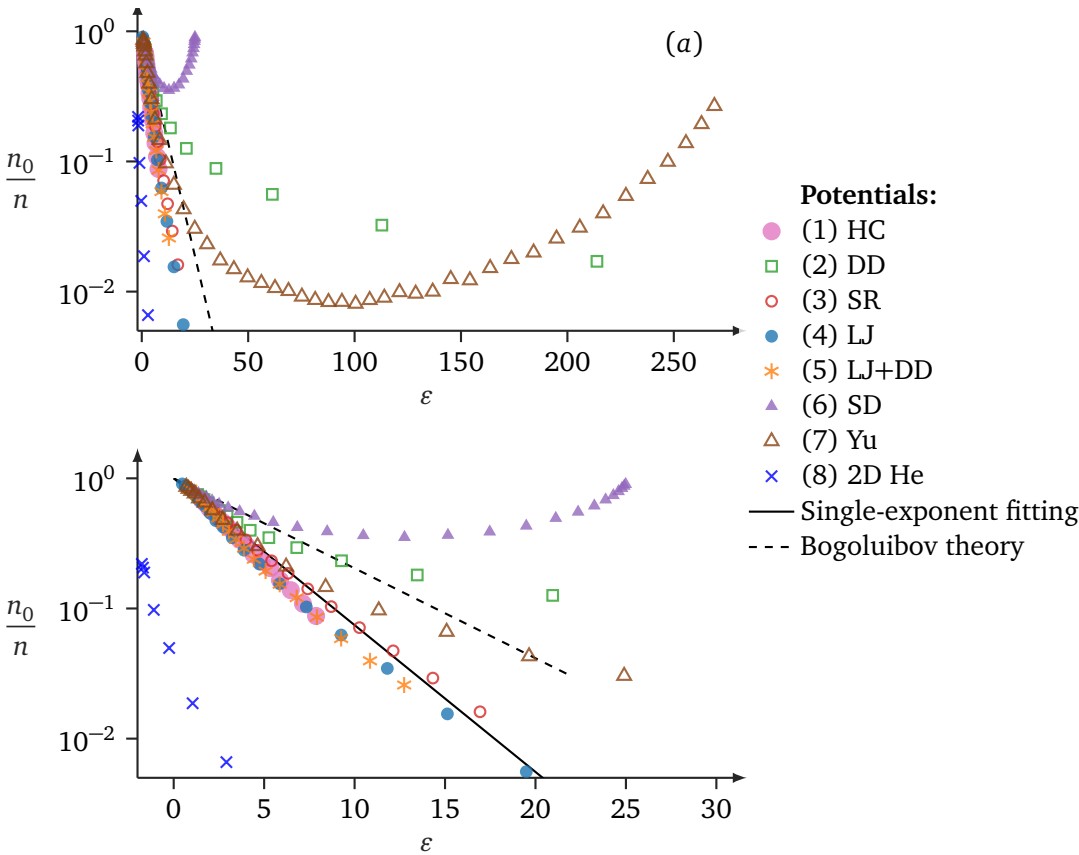

Figure 1: Condensate fraction $n_0/n$ as a function of the scaled total energy $\varepsilon = mE/(\hbar^2 Nn)$ shown on a semi-logarithmic scale on small (a) and large (b) scales. Symbols represent Monte Carlo results. The dashed line shows the prediction (8) of the Bogoliubov theory applicable in $\varepsilon \ll 1$ limit. In the panel (b) the linear regression is plotted in black.

To test the validity of the relation (8), we summarize in Fig. 1 the numerical results obtained for all considered potentials. The plot reports the condensate fraction as a function of the normalized energy $\varepsilon$. Exponential decay (8) obtained in the perturbative regime, $\varepsilon \ll 1$, corresponds to a straight line on a semilogarithmic scale. Interestingly, for $\varepsilon \gtrsim 1$ the decay is also roughly exponential for short-range potentials, although the slope is different.

We fit the data for $\ell \in \{1, 3, 4, 5\}$ potentials in the whole range of $\varepsilon$ presented in Fig. 1 (b) with a constant-exponent decay,

$$\frac{n_0}{n} \approx e^{-0.273\varepsilon} = \exp\left(-\frac{\varepsilon}{1.166\pi}\right). \tag{9}$$

The obtained fit is shown with a solid line and effectively captures the main characteristics of the decay.

Notably, for two of the "soft-core" potentials ($U_6(r)$ and $U_7(r)$) we observe non-monotonic dependence. The increase in condensate fraction for higher energies is physically reasonable. For a Bose system with a particle density $n$ high enough to effectively screen the long-range "tail" of interparticle interaction potential (this is the case for $nr_0^2 \gg 1$), only the integrable (at $r \to 0$) short-range core affects the condensate fraction. In this regime, Bogoliubov theory is again applicable (though the condensate fraction is not universally dependent on scattering length and density for that case).

In the following Sections, we will improve further the description to explain deviations from the exponential law for long-range potentials $l \in \{2, 6, 7, 8\}$.

## 3.2 Constant-exponent relation based on the quantum energy

A common feature of all the cases where exponential law is not valid is the slow convergence of the potential energy. It can be argued that if a certain relation between the condensate fraction and the energy exists, it should not include the total potential energy for long-range potentials. Indeed, the discrepancy is most dramatic in the case of the Coulomb interaction for which the total energy diverges while the condensate fraction, evidently, remains finite. The standard way to remove the divergence from the potential energy in the Coulomb case is to subtract the potential energy of interaction with a background of the opposite sign charge ("jellium" model). In a similar fashion, we propose to subtract from the total energy the potential energy of a perfect crystal

$$E^{\mathrm{cls}} \equiv \frac{N}{2} \sum_{i>1} U(r_i^0 - r_1^0), \tag{10}$$

where $r_i^0$ represents the equilibrium positions in the $T = 0$ classical crystal. The summation here is over the rest ($i \neq 1$) of $N$ particles as well as an infinite set of images of each of $N$ particles. Contribution (10) will be further referred to as the "classical energy" and the difference to the total energy as the "quantum energy", denoted by $E^{\mathrm{qnt}} \equiv E - E^{\mathrm{cls}}$. It is important to note that the uncorrelated approximation, in which the summation in Eq. (10) is replaced by an integral, $E^{\mathrm{cls}} \propto \int_0^\infty U(\mathbf{r})d\mathbf{r}$, is invalid due to a possible interaction potential divergence at $r = 0$ (interaction of hard cores being an example). In our calculations we explicitly evaluate the sum (10) for the triangular lattice which commonly is the energetically preferable packing in two dimensions. However, one should be aware that this is not the only possible option and other lattice packings are principally possible in 2D classical systems [47].

In order to improve the accuracy of the single-exponent relations (8,9) between the energy and the condensate fraction in the case of the long-range potentials, we substitute the reduced total energy $\varepsilon$ by the reduced quantum energy, defined as

$$\mathscr{E}^{\mathrm{qnt}} = \frac{m}{\hbar^2} \frac{E^{\mathrm{qnt}}}{Nn} = \frac{m}{\hbar^2} \frac{E - E^{\mathrm{cls}}}{Nn} = \varepsilon - \frac{m}{\hbar^2} \frac{E^{\mathrm{cls}}}{Nn}. \tag{11}$$

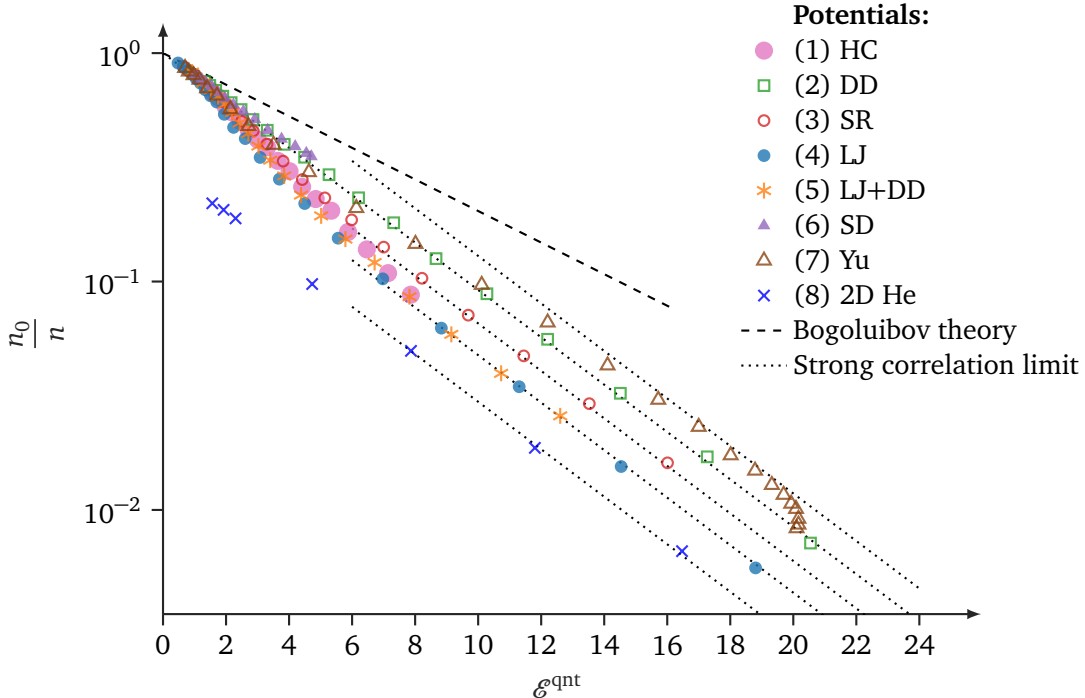

Figure 2: Condensate fraction dependence on the scaled quantum energy. Notation is similar to the one in Fig. 1. The dashed line shows the prediction of the Bogoliubov theory while the dotted lines correspond to the fits performed in the regime of strong correlations.

The resulting dependence is reported in Fig. 2 revealing two distinct regimes. In the weakly correlated regime, characterized by low quantum energies and high condensate fractions, the BT is still applicable due to the negligible difference between the full and quantum energies. For higher quantum energies, in the strongly correlated regime, the data for all the potentials demonstrates exponential decay of condensate fraction with a remarkably similar exponent being shared:

$$\frac{n_0}{n} = \exp\left(A_\ell - \frac{\mathscr{E}^{\mathrm{qnt}}}{\eta_\ell \pi}\right) \tag{12}$$

with $\eta_\ell \approx 1.35$.

In fact, the exponential dependence on quantum energy may be reasonably well described by drawing an analogy with the Bose-Hubbard model and using harmonic crystal approximation for the strongly correlated state of the Bose system at $\mathscr{E} \gg 1$. In Appendix A we argue that the condensate fraction may be expressed in terms of the Lindemann ratio $\gamma_L$ for the quantum 2D crystal at $T = 0$:

$$\frac{n_0}{n} \propto e^{-\frac{1}{4}\gamma_L^2} \tag{13}$$

and draw a connection with the quantum energy, which leads to a close value of $\eta_\ell \approx 1.4$.

In the scope of the harmonic crystal model, one may identify the quantum energy with the zero-point energy of the quantum crystal, which clearly implies its equipartition between potential and kinetic contributions, i.e.

$$\mathscr{T} = \frac{\mathscr{E}^{\mathrm{qnt}}}{2} \tag{14}$$

where $\mathscr{T} = mE^{\text{kin}}/(N\hbar^2 n)$ is the dimensionless kinetic energy.

To check the validity of the equipartion relation (14), in Fig. 3 we plot kinetic energy $\mathscr{T}$ versus the quantum energy $\mathscr{E}^{\text{qnt}}$ as given by the Monte-Carlo simulation results. First, it can be noted that for systems that stay in the gas phase, the low-density limit corresponds to vanishing of both the kinetic and quantum energies, while for a liquid state (like the liquid helium case), low density regime is not accessible and small quantum energy is achieved at a finite density which also corresponds to a finite value of the kinetic energy.

Secondly, the equipartition relation is not generally satisfied as it is evident from the deviations of the scatter points from the dashed line. Although, for large values of $\mathscr{E}^{\text{qnt}}$, the dependence is close to linear (for some of the potentials the linear fitting results are presented). One may see deviations for both the slope and the intercept, although the latter are more significant. As we discuss in B, these deviations may be largely attributed to anharmonic contribution to the effective trapping potential for a single particle created by the others.

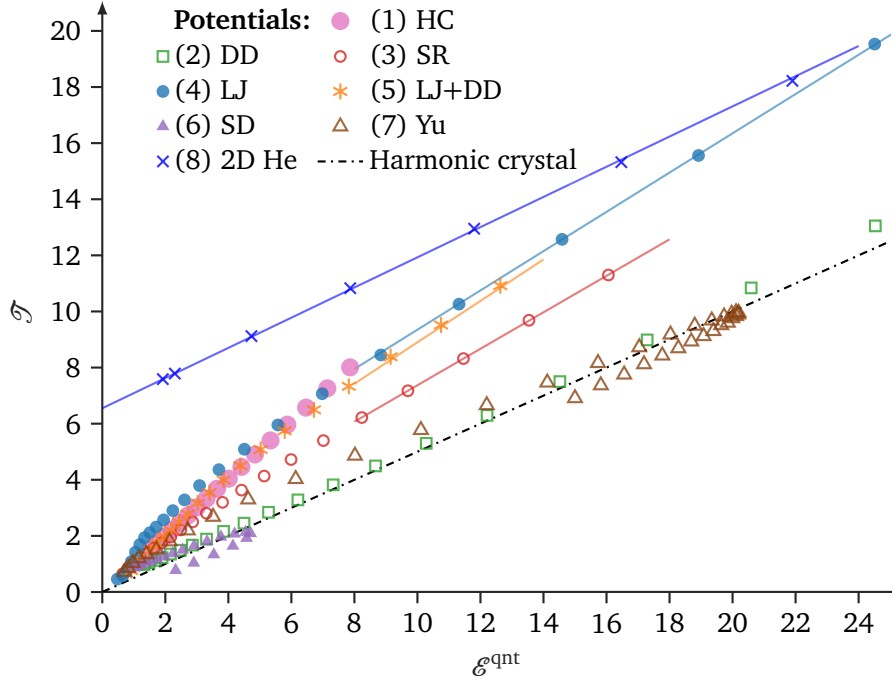

Figure 3: Kinetic energy plotted against quantum energy for all considered potentials, computed using Quantum Monte Carlo. The dash-dotted line represents the equipartition relation $\mathscr{T} = \mathscr{E}^{\text{qnt}}/2$ predicted by the harmonic crystal model. Solid lines indicate linear fits at high $\mathscr{E}^{\text{qnt}}$ for potentials that demonstrate significant deviations from the equipartition relation.

Having analyzed both the limiting cases (the weakly correlated gas and the regime of extreme strong correlations), we set our next goal as matching these asymptotic relations in a universal manner.

From the figure it is possible to infer that the harmonic regime is not yet reached for the densities we use in numerical calculations. What we observe is a broad crossover from the Bogoliubov to the harmonic crystal regimes. However, this transition has its own universal features, which are of interest to us. Note that achieving the harmonic crystal regime might be impractical, both in numerical simulations and in experiments, since it requires an extremely low condensate fraction at $T = 0$.

### 3.3 Variable-exponent relation based on the quantum energy

While both the weakly and strongly correlated regimes are reasonably well described by single-exponent relations (8) and (12), the latter has coefficients that depend on the specific potential under consideration. This implies that a proper expression for condensate fraction should include an additional variable except the quantum energy $\mathcal{E}^{\mathrm{qnt}}$. We propose using the rescaled kinetic energy $\mathcal{T}$, as introduced before, to both describe the strongly correlated regime and to interpolate between (8) and (12).

Both goals are achieved by using the following implicit expression:

$$\frac{n_0}{n} = \exp\left(-\frac{\mathcal{E}^{\mathrm{qnt}}}{\eta \pi}\right) \tag{15}$$

where the decay constant $\eta$ now is a function of the system's parameters $n_0/n$, $\mathcal{E}^{\mathrm{qnt}}$, etc. More specifically, we propose the following functional form for the decay energy $\eta$,

$$\eta = 2 - \frac{2+\kappa^2}{\pi}\left[1 - \left(\frac{n_0}{n}\right)^{\gamma}\right], \tag{16}$$

in terms of the condensate fraction itself and a coefficient $\kappa$

$$\kappa = \frac{\mathcal{E}^{\mathrm{qnt}}}{\mathcal{T}} - 2, \tag{17}$$

which, in turn, depends on the ratio between the kinetic-to-quantum energy ratio $\mathcal{T}/\mathcal{E}^{\mathrm{qnt}}$. The numerical coefficient $\gamma$ was obtained in a fitting procedure, which led to $\gamma \approx 3.37$.

This interpolating expression should properly reproduce both the limiting cases. Indeed,

- it properly captures the Bogoliubov theory prediction. Namely, for $n_0/n \to 1$, at low densities, the classical energy vanishes (since all the potentials under consideration are integrable at $\infty$), $\eta = 2$ and, consequently

$$\frac{n_0}{n} = \exp\left(-\frac{\varepsilon}{2\pi}\right), \tag{18}$$

  as suggested by (5).

- In the opposite limit we consider (15) with substitution $n_0/n = 0$ in the right-hand-side. Using the linear dependence $\mathcal{T} = \alpha \mathcal{E}^{\mathrm{qnt}} + \beta$ as implies the graph 3, we obtain:

$$\frac{n_0}{n} = \exp\left(-\frac{\mathcal{E}^{\mathrm{qnt}}}{2\pi - 2 - \frac{1}{\pi}\left(\frac{\mathcal{E}^{\mathrm{qnt}}}{\alpha \mathcal{E}^{\mathrm{qnt}} + \beta} - 2\right)^2}\right). \tag{19}$$

The exponent may be expanded as follows:

$$\ln\left(\frac{n_0}{n}\right) = -\frac{4\pi\alpha\beta\left(\alpha - \frac{1}{2}\right)}{\left[(\pi^2 - \pi - 2)\alpha^2 + 4\alpha - 1\right]^2} - \frac{\pi}{2\pi(\pi - 1) - \left(\frac{1}{\alpha} - 2\right)^2}\mathcal{E}^{\mathrm{qnt}} + O\left(\frac{1}{\mathcal{E}^{\mathrm{qnt}}}\right). \tag{20}$$

This properly reproduces the strong-coupling limit in the form (12) with

$$\eta_\ell = \frac{1}{\pi^2}\left[2\pi(\pi - 1) + \left(\frac{1}{\alpha} - 2\right)^2\right]. \tag{21}$$

Equations (15-17) establish a relationship among three physical quantities: condensate fraction, quantum and kinetic energies. That relation implicitly defines the condensate fraction as a function of the quantum and kinetic energies. While in the standard BT, both the condensate fraction and total energy depend solely on the gas parameter, resulting in the condensate fraction being implicitly dependent on a single parameter (the total energy), our new approach goes a step further by introducing a second parameter (kinetic energy). This extra parameter effectively introduces the model-specific corrections.

## 4 Universality

When discussing the universality of the established relation, we separately consider potentials $\ell = 1$ to 6 (for these potentials, we always deal with a gaseous phase in simulations) and then focus on a system of He atoms.

**Gaseous phase**

Equations (15-17) establish an empirical relation between energetic properties (kinetic energy and total energy with the perfect lattice energy excluded) and non-local quantities (condensate fraction, occupation of $k = 0$ state in the momentum distribution). To validate the proposed relation, we show in Fig. 4 the resulting prediction for the condensate fraction (solid lines) and the exact condensate fraction (symbols) obtained in Monte Carlo simulations. Remarkably, excellent agreement is observed for all considered potentials, independently of their short- or long-range nature. Furthermore, the remarkable consistency spans over many orders of magnitude of the gas parameter, ranging from the weakly interacting regime, where the depletion is small, to the strongly correlated regime, where the condensate fraction is substantially decreased or is even vanishingly small. For some potentials the gas parameter is large, $na^2 \gg 1$, and this regime lies beyond the validity of standard perturbative theories, which are typically applicable only when $na^2 \ll 1$. For the two "soft-core" potentials $U_{6/7}$ we omit data points in the high-density weak correlation regime because of the elevated level of numerical errors (see also Sec. 4.2).

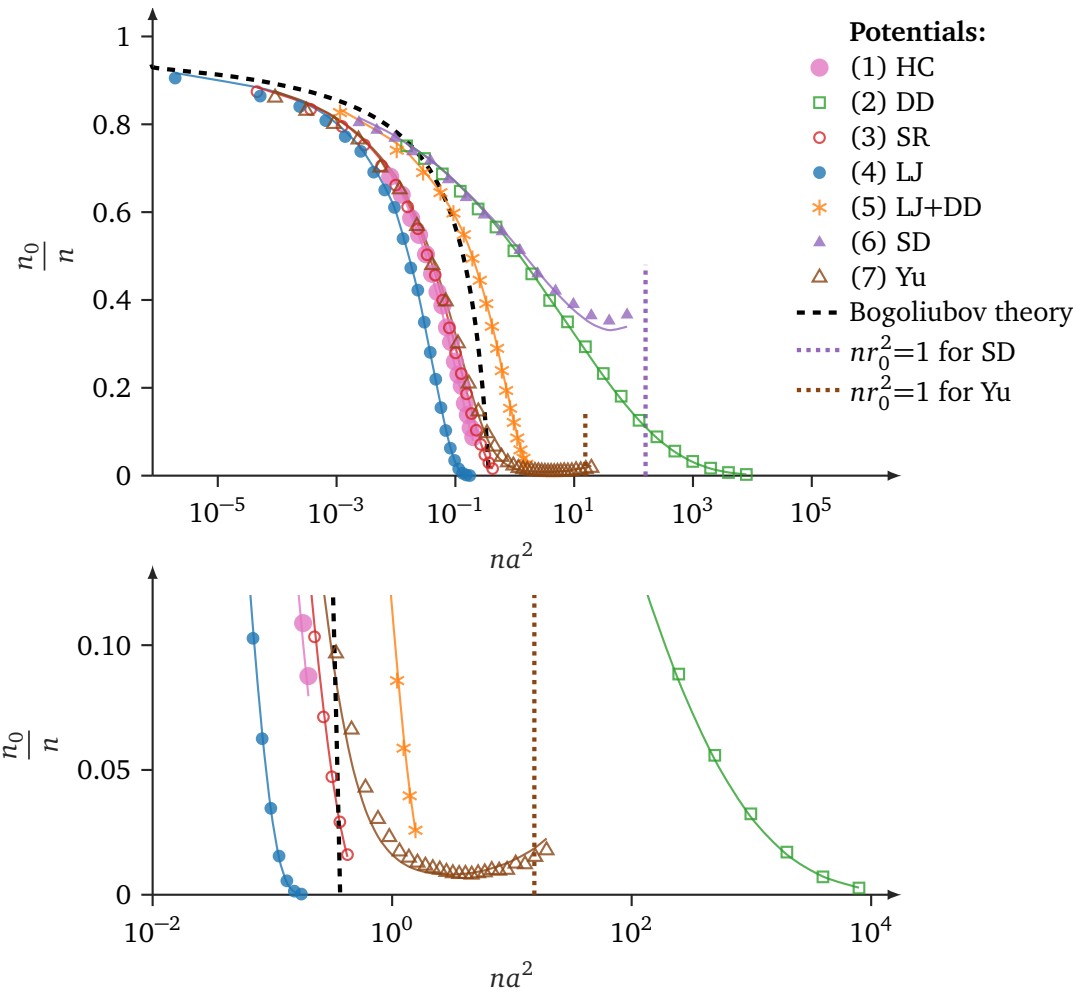

Figure 4: Condensate fraction $n_0/n$ as a function of the gas parameter $na^2$, where $a$ is the $s$-wave scattering length. Solid lines represent the predictions (15–17) of our phenomenological theory, while symbols correspond to Monte Carlo simulation results. The dashed line shows the prediction (3) of the Bogoliubov theory, which is asymptotically valid in the limit $na^2 \to 0$. The two dotted vertical lines indicate the densities at which the interparticle distance equals the potential spatial range $r_0$ for the two soft-core potentials. As discussed in Section 3.1, the gas becomes weakly correlated for $nr_0^2 \gg 1$.

It becomes evident from Fig. 4 that deep in the dilute limit, $na^2 \to 0$, the condensate fraction converges to the perturbative result (3) predicted by the Bogoliubov theory.

We claim that Eqs. (15-17) constitute a universal relation between the condensate fraction, quantum, and kinetic energies, such that the exact shape of the interparticle interaction is irrelevant. The apparent visual agreement observed in Fig. 4 is supported by examination of the data points, which show that for all the potentials fitting residuals $\Delta n_0/n$ do not exceed 0.02.

The residuals, quantifying the differences between predicted and exact values, are notably small even in the regime of strong correlations. Moreover, our theory proves to be robust as applied to different potentials resulting in consistently small deviations. This supports our claim regarding the universality of the established relation, as well as demonstrates the good overall quality of the fit.

## 4.1  Liquid Helium

We would like to focus specifically on testing our theory on the paradigmatic example of strongly-correlated quantum systems − liquid helium [48–54]. This system is known to be notoriously difficult to describe using perturbative approaches as the mean interparticle distance always remains of the order of the potential range, $\sigma = 2.556$ Å. Indeed, the dimensionless density of the liquid helium spans from $n\sigma^2 = 0.228(2)$ at the spinodal point [50] up to $n\sigma^2 = 0.443 - 0.471$ at the melting-freezing transition [48], so that the parameter $n\sigma^2$ is always of the order of unity and cannot be perturbatively small. Furthermore, the the value of the $s$-wave scattering length $a_{3D} = 104$ Å [55] is huge compared both to the range of the potential and the mean interparticle distance, so that the actual value of the $s$-wave scattering length is irrelevant in determining the properties of the liquid helium, making it very different from the usual situation found in ultradilute gases. Thus, the study of liquid helium becomes a stringent test of our results.

   In Fig. 5 (a) we report the comparison of the condensate fraction as obtained by using the phenomenological expression and confronted with the results of the Monte Carlo simulation. We find a good agreement for densities $n\sigma^2 \gtrsim 0.3$ up to the largest density at which the helium solidifies. It might seem counterintuitive that our theory works well at high densities, where the system is extremely correlated, and instead it results in larger deviations at smaller densities. The reason is that the system under consideration stays in the liquid phase and not in the gas one. The region close to the spinodal point has negative pressure, which is not possible in the gas phase.

   Intuitive argument behind the decrease of the fitting residuals observed in 5 (b) is as follows: the functional relation (8) between the condensate fraction and the total energy which is the starting point of our reasoning, $\ln(n_0/n) \propto -E$, fails completely for $E < 0$ as it produces nonphysical result of the condensate fraction being larger than unity. Indeed, the total energy in the vicinity of the equilibrium density of the liquid is negative. However, as shown in Fig. 5 (c), the total energy in a liquid becomes positive at a large density. In this way, functional relation (8) once again becomes a good starting point, resulting in a good agreement of the final expression with simulation results.

## 4.2  Applicability limits

As the results presented in this section demonstrate, our phenomenological expression shows very good agreement with Monte Carlo data across all interaction potentials and over a wide range of densities. Nevertheless, it is important to clearly understand the limits of applicability of our formula.

   Firstly, we deliberately excluded high-density data ($nr_0 \gg 1$) for soft-core potentials $U_{6/7}(r)$ due to two main concerns. In this regime, the kinetic energy is much smaller than the total and potential energies, making it sensitive to numerical inaccuracies. Moreover, significant dependence on the number of particles indicates strong finite-size effects. These limitations prevent us from drawing reliable conclusions about the universality of our formula in this region.

   In addition, the universal relation is not tested in the region close to the liquid-gas phase transition. This is due to complications arising with the macroscopic limit extrapolation [40]. Note that even for helium, we do not approach the spinodal point $n\sigma^2 \approx 0.23$ (the dotted vertical line on Fig. 5 (c)).

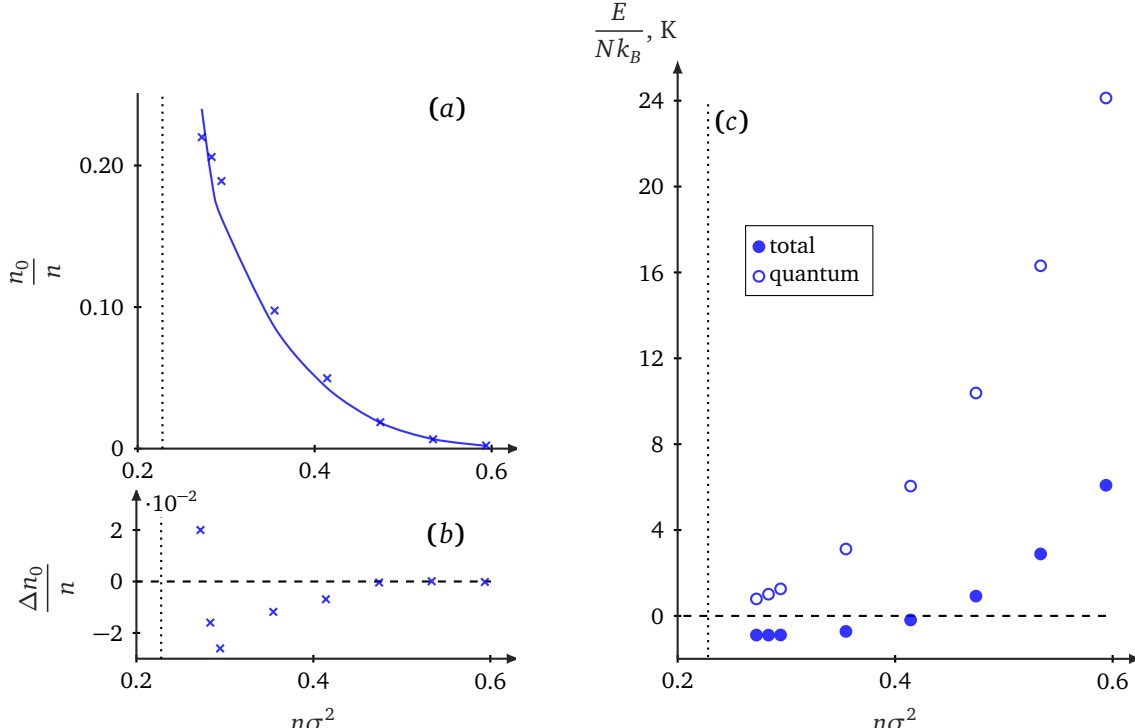

Figure 5: (a) Condensate fraction for 2D liquid helium as a function of $n\sigma^2$ with $\sigma = 2.556$ Å being the characteristic length of the interaction potential. (b) Fitting residuals as a function of density for the same system; (c) Energy per particle $E/N$. Solid symbols depict total energy $E$, hollow symbols — the quantum energy, defined as $E - E_{\mathrm{cls}}$, see Eq. (10).

## 5   Conclusions

We perform detailed Monte Carlo simulations for Bose gases with interaction potentials of various functional forms in a wide range of densities at zero temperature. Guided by analytical results for dilute and strongly correlated systems, we introduce notions of classical (10) and quantum (11) energies and propose an analytic relation (15-17) between the condensate fraction, quantum and kinetic energies. This relation constitutes the main result of our study. We provide numerical evidence for the excellent quality of the proposed phenomenological expressions in a wide range of densities. Also, we explicitly specify the region of applicability of the established relation.

The universality we point out in the paper is a phenomenological one. It describes reasonably well the transition from the Bogoliubov regime to the regime of extremely strong correlations that completely deplete the condensate. While the final fitting expression is not derived from strict reasoning and leaves some room for further studies, its very existence is a strong argument in favor of the universal behavior. Given the independence of the discovered relation on the microscopic details of the interaction potential, we expect it to hold in multiple experimentally accessible Bose systems such as ultracold atoms in geometries of reduced dimensionality and excitons in TMDC layers.

# Acknowledgements

G.E. Astrakharchik acknowledges support by the Spanish Ministerio de Ciencia e Innovación (MCIN/AEI/10.13039/ 501100011033, grant PID2020-113565GB-C21), and by the Generalitat de Catalunya (grant 2021 SGR 01411). N.A. Asriyan, I.L. Kurbakov and Yu.E. Lozovik acknowledge the support by the Russian Science Foundation grant No. 23- 42-10010, https://rscf.ru/en/project/23-42-10010/. Numerical calculations are performed by I. L. Kurbakov in the scope of the project FFUU-2024-0003.

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

# A   Condensate fraction for the strongly correlated regime

In a strongly interacting regime, the system of bosons exhibits the emergence of short-range order. The condensate fraction $n_0/n$ becomes small, which is seen as a small value of the off-diagonal long-range order,

$$\frac{n_0}{n} = \frac{1}{N} \int d^2 r \underbrace{\lim_{\delta r \to \infty} \langle \hat{\psi}^\dagger(r)\hat{\psi}(r+\delta r)\rangle}_{\rho_1(\infty)}, \tag{22}$$

where $N$ is the total particle number. For a strongly correlated particle system, the one-body density matrix $\rho_1(r)$ is typically an oscillatory function with its first minimum being located not far from the mean interparticle distance $a$. Assuming that $\rho_1(a)$ is of the same order as $\rho_1(\infty)$ (which is confirmed by numerical simulations for the potentials under consideration), we may use an approximation:

$$\frac{n_0}{n} \sim \frac{1}{N} \int d^2 r \rho_1(a). \tag{23}$$

We observed that, in the strongly correlated regime, snapshots of particle coordinates obtained in Monte Carlo simulations typically show a crystalline-like short-range order. This motivates us to expand the field operators in terms of Wannier states, which leads to the following expression:

$$\frac{n_0}{n} \sim \sum_{j=1}^{N_{\rm nn}} \int d^2 r \, w^*(r) w(r + a_j). \tag{24}$$

Here $w(r)$ is the Wannier function localized at $r = 0$, $j$ enumerates the $N_{\rm nn}$ nearest neighbors of the particle at the origin in the quasistatic configuration with $a_j$ being the vector giving the distance and direction to $j^{\rm th}$ neighbor. The Wannier functions are defined analogously to the ones of the Bose-Hubbard model (see [56]) with the effective trapping potential $V(r)$ created by other particles instead of the optical lattice. Note that we may introduce a single Wannier function for all the configuration sites even without short-range order, only since the main contribution to the potential is from the neighboring particles.

Assuming small overlap of the Wannier functions localized at the neighboring sites of the quasistatic configuration (i.e. large degree of localization) in the effective trapping potential $V(r)$, we may approximate them with Gaussians (motivated by the harmonic form of the effective trapping potential for small displacements):

$$\frac{n_0}{n} \sim \sum_{j=1}^{N_{\rm nn}} \int d^2 r \, \frac{1}{2\pi\sigma^2} \exp\left(-\frac{r^2}{2\sigma^2}\right) \exp\left(-\frac{(r+a_j)^2}{2\sigma^2}\right) \sim \exp\left(-\frac{a^2}{4\sigma^2}\right) = e^{-\frac{1}{4}\gamma_L^{-2}}. \tag{25}$$

Here $\sigma$ stands for the mean squared displacement of each particle from the effective trapping potential minimum and $\gamma_L = \sigma/a$ is the Lindemann ratio. This expression may be transformed to a form similar to (15) with

$$\eta = \frac{4}{\pi} \mathscr{E}^{\rm qnt} \gamma_L^2. \tag{26}$$

For the strongly correlated $T = 0$ system under consideration, the quantum energy $\mathscr{E}$ may be identified as the zero-point energy of the 2D crystal. In the same framework one may evaluate the Lindemann ratio. Assuming small displacements from the equilibrium position, we

employ the harmonic crystal model and consider a 2D triangular lattice of particles with nearest neighbor interaction via one of the potentials from Table reftab:potentials. The interaction $U_{ij}$ between the adjacent sites $i$ and $j$ may be expanded in the vicinity of the lattice constant $a$ up to the second order in displacements $\boldsymbol{u}_{i(j)}$:

$$U_{ij} - U(a) \approx U'(a)(|\boldsymbol{r}_{ij} + \boldsymbol{u}_j - \boldsymbol{u}_i| - a) + \frac{U''(a)}{2}(|\boldsymbol{r}_{ij} + \boldsymbol{u}_j - \boldsymbol{u}_i| - a)^2 =$$

$$= -f(\boldsymbol{n}_{ij} \cdot \Delta\boldsymbol{u}_{ij}) + \frac{\kappa}{2}(\boldsymbol{n}_{ij} \cdot \Delta\boldsymbol{u}_{ij})^2 - \frac{f}{2a}\left[(\Delta\boldsymbol{u}_{ij})^2 - (\boldsymbol{n}_{ij} \cdot \Delta\boldsymbol{u}_{ij})^2\right]. \qquad (27)$$

Here $\boldsymbol{n}_{ji} = \boldsymbol{r}_{ij}/a = (\boldsymbol{r}_j - \boldsymbol{r}_i)/a$, $\Delta\boldsymbol{u}_{ij} = \boldsymbol{u}_j - \boldsymbol{u}_i$ and $f \neq 0$ since with externally fixed density (which is the case for the simulations described in this paper) the system may be in a non-equilibrium "squeezed" state.

Diagonalization of the dynamic matrix for a triangular lattice leads to the following phonon spectrum of the standard form:

$$\omega_{\|/\perp}^2(\boldsymbol{k}) = \frac{\kappa}{m}\left\{(1-\beta)\left[3 - \cos(ak_x) - \cos\left(\frac{a}{2}(k_x - \sqrt{3}k_y)\right) - \cos\left(\frac{a}{2}(k_x + \sqrt{3}k_y)\right)\right]\right.$$

$$\pm \frac{1+\beta}{\sqrt{2}}\left[3 + \cos^2(ak_x) + \cos\left(\frac{ak_x}{2}\right)\cos\left(\frac{\sqrt{3}ak_y}{2}\right) - 2\cos^3\left(\frac{ak_x}{2}\right)\cos\left(\frac{\sqrt{3}ak_y}{2}\right)\right.$$

$$\left.\left. - \cos(\sqrt{3}ak_y) + \cos(ak_x)\left(2\cos(\sqrt{3}ak_y) - 1 - 3\cos\left(\frac{ak_x}{2}\right)\cos\left(\frac{\sqrt{3}ak_y}{2}\right)\right) - \sin^2(ak_x)\right]^{1/2}\right\}.$$

$$(28)$$

Here, the dimensionless number $\beta = f/(\kappa a)$ quantifies the "squeezing degree" of the system. The $\pm$ signs correspond to longitudinal and transverse phonons with the following sound velocities:

$$v_\perp = a\sqrt{\frac{3\kappa}{8m}(1 - 3\beta)}, \; v_\| = a\sqrt{\frac{9\kappa}{8m}\left(1 - \frac{\beta}{3}\right)}. \qquad (29)$$

Reasonably, strong squeezing of the crystal with $\beta > 1/3$ leads to diverging transverse quantum fluctuations which destroy the crystal.

With the phonon spectrum given, one may evaluate the desired constant

$$\eta = \frac{4}{\pi}\mathscr{E}^{\text{qnt}}\gamma_L^2 = \frac{4}{\pi}\left[\frac{m}{\hbar^2 n}\frac{1}{n}\int\limits_{\boldsymbol{k}\in\text{BZ}}\frac{d^2\boldsymbol{k}}{(2\pi)^2}\frac{\hbar\omega_\perp(\boldsymbol{k}) + \hbar\omega_\|(\boldsymbol{k})}{2}\right]\left[\frac{1}{na^2}\int\limits_{\boldsymbol{k}\in\text{BZ}}\frac{d^2\boldsymbol{k}}{(2\pi)^2}\left[\frac{\hbar}{2m\omega_\perp(\boldsymbol{k})} + \frac{\hbar}{2m\omega_\|(\boldsymbol{k})}\right]\right].$$

$$(30)$$

The integration of the functions of $\omega_{\|/\perp}$ over the hexagonal Brillouin zone may be performed numerically and leads to the results presented in Fig.6 as a function of $\beta$.

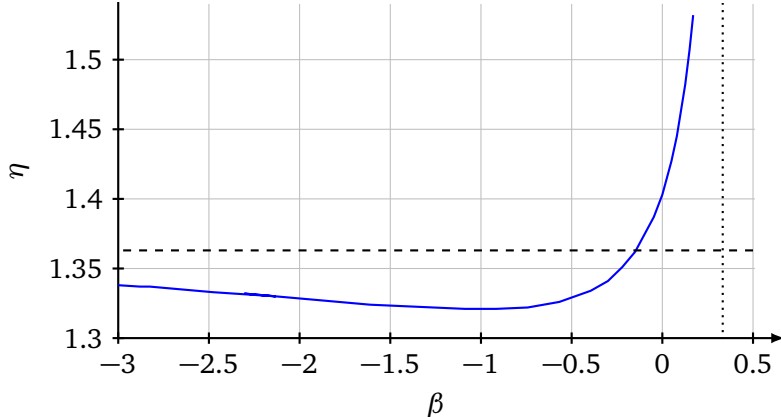

Figure 6: The results of numerical calculation of the parameter $\eta$ from (15) with the help of (30). The dotted vertical line $\beta = 1/3$ shows the critical squeezing value that leads to crystal melting due to zero-point fluctuations. The dashed horizontal line $\eta \approx 1.363$ corresponds to the value extracted from the Monte-Carlo simulations (Fig. 2).

Note that the values of $\eta$ are reasonably close to the ones extracted from the semi-log plot in Fig. 2 for $\mathscr{E}^{\mathrm{qnt}} \gg 1$. However, we are not able to evaluate the exact pre-exponential factor, since, as qualitatively demonstrated in Appendix B, it is very sensitive to the anharmonic contributions to the interaction (phonon interaction). In addition, the harmonic crystal model predicts slight potential-dependence of the parameter $\eta$ as may be inferred from the Fig. 6 and the table 2 below. The limited accuracy of the simulation results in the strongly correlated regime prevents numerical verification of this dependence.

| $\ell$ | $U_\ell(r)$ | Description | $f$ | $\kappa$ | $\beta$ |
|---|---|---|---|---|---|
| 2-3 | $\dfrac{\hbar^2}{mr_0^3}\dfrac{r_0^k}{r^k}$ | Power-law | $> 0$ | $> 0$ | $\frac{1}{1+k}$ |
| 4 | $\dfrac{\hbar^2 U_0}{mr_0^2}\left(\dfrac{r_0^{12}}{r^{12}} - \dfrac{r_0^6}{r^6}\right)$ | LJ | $\propto 2-\left(\frac{a}{r_0}\right)^6$ | $\propto 26-7\left(\frac{a}{r_0}\right)^6$ | $\dfrac{\left(\frac{a}{r_0}\right)^6 - 2}{7\left(\frac{a}{r_0}\right)^6 - 26}$ |
| 7 | $\dfrac{U_0}{mr_0 r}\exp\left(-\dfrac{r}{r_0}\right)$ | Yu | $> 0$ | $> 0$ | $\dfrac{1+a/r_0}{2+2(a/r_0)+2(a/r_0)^2}$ |

Table 2: Squeezing parameter $\beta$ evaluated for some of the potentials under consideration

# B The impact of quartic perturbation to the harmonic model

When discussing the dependence of the kinetic energy on the quantum energy in the main text, we witnessed deviations from the equipartition relation. In this appendix, we discuss some reasons for such behavior.

We consider a triangular 2D lattice with lattice constant $a$, composed of particles interacting via one of the potentials $U_\ell(r)$. In the spirit of the Einstein model we consider a single particle at $\boldsymbol{r} = 0$ in a potential, created by all the others ($\boldsymbol{u}$ is the deviation from the equilib-

rium position):

$$V(\boldsymbol{u}) = \sum_i U(\boldsymbol{r}_i - \boldsymbol{u}) = V_0 + \frac{V''}{2}u^2 + \frac{V^{(iv)}}{4!}u^4 + ... = V_0 + \frac{\hbar\omega}{2}\frac{u^2}{l^2} + \frac{\alpha}{24}\frac{u^4}{l^4} + ... \tag{31}$$

Here $m$ is the particle mass, $\omega$ is the harmonic oscillator frequency and $l = \sqrt{h/(m\omega)}$ is the characteristic oscillator length scale. Note that the effective potential $V(\boldsymbol{u})$ is the same for all the particles in the perfect triangular lattice.

For a particle in this type of potential, one may use the time-independent perturbation theory in a standard manner for the single-particle Hamiltonian:

$$\hat{h} = \hat{h}_0 + \hat{v} = \hbar\omega\left(1 + \hat{a}_x^\dagger\hat{a}_x + \hat{a}_y^\dagger\hat{a}_y\right) + \frac{\alpha}{96}\left[(\hat{a}_x + \hat{a}_x^\dagger)^2 + (\hat{a}_y + \hat{a}_y^\dagger)^2\right]^2. \tag{32}$$

One may derive straightforwardly the first-order correction to the vacuum state:

$$|0,0\rangle^{(1)} = -\frac{1}{96}\frac{\alpha}{\hbar\omega}\left[4\sqrt{2}|2,0\rangle + |2,2\rangle + 4\sqrt{2}|0,2\rangle + \frac{\sqrt{6}}{2}|4,0\rangle + \frac{\sqrt{6}}{2}|0,4\rangle\right], \tag{33}$$

to the total energy of the particle:

$$e^{tot} = \hbar\omega + \frac{\alpha}{12}, \tag{34}$$

and the correction to the kinetic energy:

$$e^{kin} = \left(\langle 0,0| + {}^{(1)}\langle 0,0|\right)\hat{t}\left(|0,0\rangle + |0,0\rangle^{(1)}\right) = \frac{\hbar\omega}{2} + \frac{\alpha}{12}. \tag{35}$$

Passing to the normalized energies, we end up with the following relation:

$$\mathcal{T} = \frac{1}{2}\mathcal{E}^{qnt} + \frac{1}{24}\frac{\alpha m}{\hbar^2 n} = \frac{\mathcal{E}^{qnt}}{2} + \frac{1}{24}\frac{V^{(iv)}m}{\hbar^2 n}\frac{\hbar^2}{m^2\omega^2} = \frac{\mathcal{E}^{qnt}}{2} + \frac{1}{24}\frac{V^{(iv)}}{V''}\frac{1}{n}. \tag{36}$$

This expression demonstrates the deviations from the equipartition relation. Moreover, for a power-law potential $U(r) \propto r^{-n}$, the derivative ratio is proportional to $a^{-2}$. Thus, since $na^2 = \text{const}$, one may argue that, at least for power-law potentials, the perturbative deviations lead to a mere vertical shift of the original linear function.

Among the potentials considered by means of Monte Carlo methods, for only three of them we could achieve high enough densities, that ensure stability of the triangular crystal lattice ($V'' > 0$) and compare the perturbation theory predictions with Monte Carlo results. The comparison is presented in Fig. 7 below.

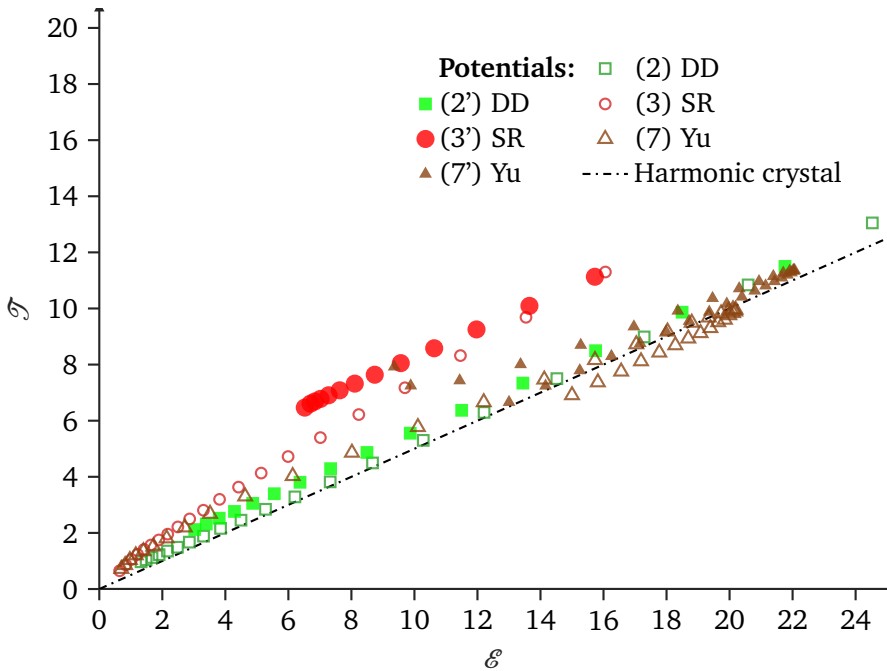

Figure 7: The Monte Carlo results for quantum and kinetic energies (hollow marks) compared with the ones evaluated by means of the perturbation theory to the harmonic crystal (filled marks, numbers with prime).

To perform the comparison, we evaluated numerically the infinite sum over all the particles in the lattice to find $V(r)$ and than evaluated $\alpha$ to use expressions (34)-(35). The results demonstrate good qualitative agreement. Thus, anharmonic terms explain significant share of the deviations from the $\mathcal{T} = \mathcal{E}^{\text{qnt}}/2$ line.