# Peer review of "A phenomenological universal expression for the condensate fraction in strongly-correlated two-dimensional Bose gases"

_SciPost Physics_

## Round 1 · Referee Report · Anonymous (Referee 1) · 2025-10-30

Strengths

1) The manuscript presents a novel universal relation among fundamental properties of quantum many-body systems, with applications on a large variety of condensed-matter systems.

2) The relation is obtained from an extensive numerical investigation based on state-of-the-art computational techniques.

Weaknesses

1) The relation is empirical, with sound theoretical foundations valid only in the asymptotic regimes.

Report

The authors present a universal relation that relates the zero-temperature condensate fraction of two-dimensional quantum many-body systems to the kinetic energy and a properly-defined quantum energy. This relation is obtained by fitting exact quantum Monte Carlo data for a variety of interatomic potentials, ranging from hard-core models, to Yukawa interaction, and to Helium atoms. This relation is shown to be applicable from the low-density to the strongly-correlated regimes, and the asymptotic limits are discussed in terms of Bogoliubov theory and a harmonic crystal model.

The relation introduced in this manuscript is original and, as one might foresee, it will find applications in different subfields of condensed matter physics. The numerical analysis is exhaustive and based on reliable computational algorithms. Therefore, I think that the manuscript is suitable for publication in SciPost Physics. Below I report a few technical questions, to be addressed before publication.

Requested changes

1) In the introduction, the authors use the same symbol for the 3D and the 2D scattering lengths. I suggest adopting different symbols to avoid confusion.

2) The authors only briefly mention the 3D scenario. In particular, I suggest to include some comments on whether an analogous relation might be derived for the 3D case, or whether this type of universality is, for some reason, peculiar to the 2D geometry.

3) The authors consider a variety of interatomic potentials, both finite range and with slower decay with distance. I suggest clearly discussing for which models the scattering length can be defined, and how exactly the authors computed the scattering length for each model.

4) The authors mention the phenomenon of crystallization in the high density regime. I suggest further expanding the discussion on the applicability of the universal relation in this regime. Specifically, is the relation supposed to break down as soon as the crystal forms? Does this relation allow one to make any conjecture on the possibility of observing supersolids, where a finite condensate fraction coexists with spatial order? 

5) I suggest showing in figure 4 the crystallization density of hard disks.

6) The fact that the triangular lattice is the equilibrium classical structure for all interatomic potentials deserves some further arguments or some references to previous literature.

7) The computation of the potential energy of the hard core interaction is not clear, since the potential is either zero or infinity.

8) I do not see the label "(b)" in figure 1.

Recommendation

Ask for minor revision

---

## Round 1 · Referee Report · Anonymous (Referee 2) · 2025-11-11

Strengths

1- The manuscript provides a new simple expression relating three relevant physical quantities in a 2D quantum gas or liquid, which will be amenable to scrutiny with other theoretical approaches and in experiments.

2- The manuscript is quite clear and combines nonperturbative Monte Carlo approaches with complementary analytical approaches that are valid in some limiting regimes.

Weaknesses

1- It is not clear how necessary the functional form of Eq. 16 is, and whether other forms could provide the same accuracy.

Report

The Authors investigate 2D bosonic gases and liquids, with various different potentials, and argue for a phenomenological relation between the condensate fraction n0 at zero temperature, the kinetic energy, and the total energy minus the potential energy of a classical crystal (quantum energy).

They first discuss how a universal relation between n0 and energy holds in the weakly interacting regime, and then investigate how this should be minimally modified in order to account for model dependence. The result is quite interesting because apparently the three aforementioned quantities are sufficient to give a reasonable agreement with all the discussed potentials in a wide range of densities. I particularly appreciate the discussion on the limitations of this approach, and the description of the analytical considerations that can be done in some limiting regimes.

I find the article to be suitable for publication in SciPost Physics, because in particular it "Opens a new pathway in an existing or a new research direction, with clear potential for multi-pronged follow-up work", given the wide applicability of the proposed phenomenological relation and its falsifiability with other theoretical or experimental studies. Some minor points should be clarified and typos corrected.

Requested changes

My main requests for clarification are: 1- On page 5, the Authors mention that potential and kinetic energies are evaluated as extrapolated estimators. Since the kinetic energy enters their phenomenological relation for the condensate fraction, could they comment on how discrepancies could arise from this? I am thing in particular of the Helium liquid case, where more refined guiding functions (for example with backflow or three-body Jastrow terms) could change the value of the extrapolated kinetic energy.

2- The Authors first extrapolate Eq 5 to an exponential form, Eq 8, and then let the decay rate be a fitted variable, in Eq 9. While the soft-disk potential 6 agrees with the Bogoliubov-inspired Eq 8 for a large range of energies, it appears that potentials 1,3,4,5 need the fitted scaling of Eq 9. I am confused here, because I would expect that the Bogoliubov-inspired formula 9 was valid for all potentials (in the gas phase) for epsilon sufficiently small, but this is not clear from Figure 1b. From the wording in the paragraph, the Authors are implying that potentials 1,3,4,5 agree with Eq. 8 for very small epsilon, and then Eq 9 sets in. I guess a third panel for Figure 1, highlighting region \epsilon \in [0,1] would clarify this aspect.

3- It is not clear how necessary the functional form of Eq. 16 is, and whether other forms could provide the same accuracy. I guess other functional forms could fulfil the same mandatory limits. Could the Authors discuss this?

4- Maybe in the conclusions the authors could briefly comment on whether analogous relations hold in 3D, and at finite temperature (but in 2D there would not be n0), and in trapped systems (where maybe their numerical approach to calculate n0 would not be sufficient).

Other minor corrections 5- On page 1, the sentence "The ground state energy in a fully condensed homogeneous system [...]" should make it clear that here the Authors are referring to a 3D system, with the 3D scattering length

6- On page 2, after equation 4, the Authors should make it clear that from now on "a" and "n" are the 2D scattering length and density, respectively.

7- "Bose-Condensates" -> "Bose-Einstein condensates"

8- On page 4, Table 1, potential 7: I guess "\hbar^2" is missing, since U0 is dimensionless and for consistency with the other potentials

9- Optionally, a figure could be added showing all used potentials as a function of r, to provide also a direct visual comparison.

10- In the legends of figures 1 and 2: "Bogoluibov"->"Bogoliubov"

11- Figure 1, panel b is missing the label "(b)"

12- After Eq 12, the fitting parameter A_l should be described and related to the different offsets in Figure 2.

13- Before Eq 14, the Authors say that "In the scope of the harmonic crystal model, one may identify the quantum energy with the zero-point energy of the quantum crystal". I think it would be clearer to add that this is due to the fact that the potential energy of the classical harmonic model is zero.

14- On page 9: "in B" -> "in Appendix B"

15- On page 13: "the the" -> "the"

16- On page 21 "Table reftab:potentials" -> "Table 1"

Recommendation

Ask for minor revision

---

## Editorial Decision

awaiting_resubmission